# Preventing microalbuminuria with benazepril, valsartan, and benazepril–valsartan combination therapy in diabetic patients with high-normal albuminuria: A prospective, randomized, open-label, blinded endpoint (PROBE) study

**Piero Ruggenenti**[1,2☯], **Monica Cortinovis**[1☯]*, **Aneliya Parvanova**[1☯], **Matias Trillini**[1], **Ilian P. Iliev**[1], **Antonio C. Bossi**[3], **Antonio Belviso**[4], **Maria C. Aparicio**[1], **Roberto Trevisan**[5], **Stefano Rota**[2], **Annalisa Perna**[1], **Tobia Peracchi**[1], **Nadia Rubis**[1], **Davide Martinetti**[1], **Silvia Prandini**[1], **Flavio Gaspari**[1], **Fabiola Carrara**[1], **Salvatore De Cosmo**[6], **Giancarlo Tonolo**[7], **Ruggero Mangili**[8], **Giuseppe Remuzzi**[1], on behalf of the VARIETY Study Organization**

**1** Department of Renal Medicine, Clinical Research Center for Rare Diseases, "Aldo e Cele Daccò": Istituto di Ricerche Farmacologiche Mario Negri IRCCS, Ranica, Bergamo, Italy, **2** Unit of Nephrology and Dialysis, Azienda Socio-Sanitaria Territoriale Papa Giovanni XXIII, Bergamo, Italy, **3** Unit of Diabetology and Metabolic Diseases, Azienda Socio-Sanitaria Territoriale Bergamo Ovest, Treviglio-Caravaggio-Romano, Bergamo, Italy, **4** Poliambulatorio extra-ospedaliero, Azienda Socio-Sanitaria Territoriale Bergamo Ovest, Brembate di Sopra, Bergamo, Italy, **5** Unit of Diabetology and Endocrinology, Azienda Socio-Sanitaria Territoriale Papa Giovanni XXIII, Bergamo, Italy, **6** Unit of Internal Medicine, Research Hospital "Casa Sollievo della Sofferenza", San Giovanni Rotondo, Foggia, Italy, **7** SC Diabetologia Aziendale ASL 2 Olbia, San Giovanni di Dio Hospital, Olbia, Italy, **8** Azienda Socio-Sanitaria Territoriale Bergamo Est, Seriate, Bergamo, Italy

☯ These authors contributed equally to this work.
* monica.cortinovis@marionegri.it

## Abstract

### Background

Angiotensin converting enzyme (ACE) inhibitors and angiotensin receptor blockers (ARBs) prevent microalbuminuria in normoalbuminuric type 2 diabetic patients. We assessed whether combined therapy with the 2 medications may prevent microalbuminuria better than ACE inhibitor or ARB monotherapy.

### Methods and findings

VARIETY was a prospective, randomized, open-label, blinded endpoint (PROBE) trial evaluating whether, at similar blood pressure (BP) control, combined therapy with benazepril (10 mg/day) and valsartan (160 mg/day) would prevent microalbuminuria more effectively than benazepril (20 mg/day) or valsartan (320 mg/day) monotherapy in 612 type 2 diabetic patients with high-normal albuminuria included between July 2007 and April 2013 by the Istituto di Ricerche Farmacologiche Mario Negri IRCCS and 8 diabetology or nephrology units

**Data Availability Statement:** Sharing of individual participant data with third parties was not specifically included in the informed consent of the study, and unrestricted diffusion of such data may pose a potential threat of revealing participants' identities, as permanent data anonymization was not carried out (patient records were instead de-identified per protocol during the data retention process). To minimize this risk, researchers who wish to inquire about access to individual participant data that underlie the results reported in this article shall submit a proposal to the Laboratory of Biostatistics of the Department of Renal Medicine of the Istituto di Ricerche Farmacologiche Mario Negri IRCCS (RenMedBiostatistics@marionegri.it). To gain access, data requestors will need to sign a data access agreement and obtain the approval of the local ethics committee.

**Funding:** This trial was sponsored by the Italian Drug Agency (AIFA, Rome, Italy) (FARM5TPY8X to PR). Novartis Italia (Varese, Italy) supplied the study drugs free of charge. The funders had no role in study design, data collection and analysis, decision to publish, or preparation of the manuscript.

**Competing interests:** I have read the journal's policy and the authors of this manuscript have the following competing interests: GR is a member of PLOS Medicine's Editorial Board. All other authors declare no competing interests.

**Abbreviations:** ACE, angiotensin converting enzyme; ARB, angiotensin receptor blocker; BENEDICT, BErgamo NEphrologic DIabetes Complications Trial; BP, blood pressure; CALM II, Candesartan and Lisinopril Microalbuminuria II; CKD-EPI, Chronic Kidney Disease Epidemiology Collaboration; CONSORT, Consolidated Statement of Reporting Trials; CRC, Clinical Research Centre; GFR, glomerular filtration rate; IQR, interquartile range; MACE, major cardiovascular event; PROBE, prospective, randomized, open-label, blinded endpoint; RAS, renin–angiotensin system; ROADMAP, Randomized Olmesartan And Diabetes Microalbuminuria Prevention; SGLT2, sodium–glucose co-transporter 2; UAE, urinary albumin excretion.

in Italy. Time to progression to microalbuminuria was the primary outcome. Analyses were intention to treat. Baseline characteristics were similar among groups. During a median [interquartile range, IQR] follow-up of 66 [42 to 83] months, 53 patients (27.0%) on combination therapy, 57 (28.1%) on benazepril, and 64 (31.8%) on valsartan reached microalbuminuria. Using an accelerated failure time model, the estimated acceleration factors were 1.410 (95% CI: 0.806 to 2.467, $P = 0.229$) for benazepril compared to combination therapy, 0.799 (95% CI: 0.422 to 1.514, $P = 0.492$) for benazepril compared to valsartan, and 1.665 (95% CI: 1.007 to 2.746, $P = 0.047$) for valsartan compared to combination therapy. Between-group differences in estimated acceleration factors were nonsignificant after adjustment for predefined confounders. BP control was similar across groups. All treatments were safe and tolerated well, with a slight excess of hyperkalemia and hypotension in the combination therapy group. The main study limitation was the lower than expected albuminuria at inclusion.

## Conclusions

Risk/benefit profile of study treatments was similar. Dual renin–angiotensin system (RAS) blockade is not recommended as compared to benazepril or valsartan monotherapy for prevention of microalbuminuria in normoalbuminuric type 2 diabetic patients.

## Trial registration

EudraCT 2006-005954-62; ClinicalTrials.gov NCT00503152.

## Author summary

### Why was this study done?

- Renin–angiotensin system (RAS) blockade with angiotensin converting enzyme (ACE) inhibitors or angiotensin receptor blockers (ARBs) prevents the onset of microalbuminuria in patients with type 2 diabetes and normoalbuminuria.

- Some studies found that ACE inhibitor and ARB combination therapy reduced urinary albumin excretion (UAE) more effectively than ACE inhibitor or ARB monotherapy in type 2 diabetic patients with microalbuminuria or macroalbuminuria. Treatment effect was, however, associated with greater blood pressure (BP) reduction.

- Whether, at comparable BP control, dual RAS inhibition with an ACE inhibitor and an ARB could be more renoprotective than either monotherapy in diabetic patients with no evidence of kidney disease is unknown.

### What did the researchers do and find?

- In this prospective, randomized, open-label, blinded endpoint (PROBE) trial, we evaluated whether, at similar BP control, combination therapy with the ACE inhibitor benazepril and the ARB valsartan would reduce the incidence of microalbuminuria more effectively than benazepril or valsartan monotherapy in 612 patients with type 2 diabetes and high-normal albuminuria.

- Secondarily, we compared the effects of the 2 monotherapies on the primary prevention of microalbuminuria in this population.

- We found that during a median follow-up of 66 months, combined treatment with half of the standard manufacturer-recommended antihypertensive doses of benazepril and valsartan had no superior effect against progression to microalbuminuria as compared to monotherapy with full recommended doses of either benazepril or valsartan.

- The protective effects of benazepril and valsartan monotherapies against progression to microalbuminuria were also similar.

- All treatments were safe and well tolerated, with a slight excess of hyperkalemia and hypotension episodes in the combination therapy group.

### What do these findings mean?

- Dual RAS blockade should not be preferred to ACE inhibitor or ARB monotherapy for the primary prevention of microalbuminuria in patients with type 2 diabetes and normoalbuminuria.

- Recent studies showing that sodium–glucose co-transporter 2 (SGLT2) inhibitors may afford substantial nephro- and cardioprotection to patients with type 2 diabetes and varying degrees of albuminuria might pave the way to novel prevention strategies based upon the integrated use of these novel medications with an ACE inhibitor or an ARB, but not with their combination.

## Introduction

In most patients with diabetes, albuminuria progressively increases over time in parallel with a progressive decline in glomerular filtration rate (GFR) [1]. Both albuminuria increase and GFR decline are a continuum [2]. However, the conventional urinary albumin excretion (UAE) thresholds of 20 or 200 μg/min in overnight urine collections define the onset of micro-albuminuria or macroalbuminuria in normo- or microalbuminuric patients, respectively. Every year, approximately 2% of patients with type 2 diabetes and normoalbuminuria progress to microalbuminuria [3], and 3% of microalbuminuric diabetic patients progress to macroal-buminuria [1,4]. With macroalbuminuria, GFR decline sharply accelerates [1], and, in about 15% to 35% of affected patients, GFR decline may eventually result in end-stage kidney disease over 3 years [5].

In addition to prelude to macroalbuminuria, microalbuminuria is also a strong and inde-pendent predictor of premature cardiovascular morbidity and mortality in patients with type 2 diabetes [6,7]. Thus, strategies to limit or even prevent the onset of microalbuminuria, and its eventual progression to macroalbuminuria, are expected to translate into long-term renal, and, conceivably, cardioprotection in this population.

The BErgamo NEphrologic DIabetes Complications Trial (BENEDICT) found that treat-ment with the angiotensin converting enzyme (ACE) inhibitor trandolapril, either as mono-therapy or combined with the calcium channel blocker verapamil, halved the incidence of

microalbuminuria over a median observation period of 3.6 years in 1,204 hypertensive patients with type 2 diabetes and normoalbuminuria compared to verapamil alone or placebo [3]. The benefit extended beyond blood pressure (BP) control and was not enhanced by add-on verapamil therapy [3]. In a similar study published 7 years later, the Randomized Olmesartan And Diabetes Microalbuminuria Prevention (ROADMAP) trial, the angiotensin receptor blocker (ARB) olmesartan delayed the onset of microalbuminuria by 23% compared to placebo in mostly hypertensive patients with type 2 diabetes and normoalbuminuria [8]. This effect appeared to be largely mediated by better control of hypertension in olmesartan-treated patients than in controls.

Evidence that ACE inhibitors and ARBs appeared to share similar effects on albuminuria paved the way to diverse studies that consistently found that ACE inhibitor and ARB combination therapy reduced albuminuria more effectively than ACE inhibitor or ARB monotherapy in type 2 diabetic subjects with microalbuminuria [9] or macroalbuminuria [10,11]. Conversely, prospective trials in advanced stages of diabetic renal disease found that combination therapy does not offer any additional nephroprotective effect over ACE inhibitor or ARB monotherapy [5,12]. These findings, however, did not rule out the possibility that dual renin–angiotensin system (RAS) inhibition could be more effective than single drug inhibition in early stages of the disease, when renal changes can still be halted or even reverted [13].

Thus, we designed the VARIETY study, a prospective, randomized, open-label, blinded endpoint (PROBE) trial primarily aimed to evaluate whether, at similar BP control, combined therapy with the ACE inhibitor benazepril and the ARB valsartan would reduce the incidence of microalbuminuria more effectively than benazepril or valsartan monotherapy in hypertensive patients with type 2 diabetes at increased risk of progression to the endpoint because of high-normal albuminuria at inclusion. Secondarily, we compared the protective effect against microalbuminuria development of the 2 monotherapies in the same population.

## Methods

### Study design and participants

This was a Phase III, multicenter, parallel PROBE trial [14] that included male and female patients aged over 40 years with type 2 diabetes (WHO criteria) and a known history of diabetes not exceeding 25 years. They had to fulfill the following selection criteria: UAE rate ≥7 and <20 μg/min in at least 2 of 3 consecutive overnight collections (high-normal albuminuria), serum creatinine concentration <1.5 mg/dL and systolic and/or diastolic BP>135/85 mm Hg or concomitant treatment with BP-lowering medications. Patients with glycated hemoglobin >11% (>97 mmol/mol); serum potassium ≥5.5 mEq/L despite diuretic therapy and optimized control of blood glucose and acid/base balance; bilateral ischemic kidney disease; primary glomerular disease or any renal or systemic disease requiring chronic immunosuppressive therapy; cancer; and any other clinical condition that could jeopardize study completion or data interpretation were excluded. We also excluded patients with drug or alcohol abuse and pregnant, lactating, or potentially childbearing women without effective contraception (see study protocol in S1 Appendix and https://clinicaltrials.gov/ct2/show/NCT00503152). Selected patients were referred to the Aldo e Cele Daccò Clinical Research Centre (CRC) for Rare Diseases of the Istituto di Ricerche Farmacologiche Mario Negri IRCCS in Bergamo (Italy) and to 8 diabetology or nephrology units, all in Italy (see VARIETY Study Organization in S2 Appendix).

The study protocol and its amendments were approved by each site's ethics committee, and written informed consent was obtained from all patients in compliance with the Declaration of Helsinki. The CRC coordinated and monitored the trial according to Good Clinical Practice

guidelines. Data were recorded locally on paper case report forms and centralized at the CRC. The study was reported according to the Consolidated Statement of Reporting Trials (CONSORT) guidelines (see CONSORT Checklist in S3 Appendix). This trial is registered with ClinicalTrial.gov number NCT00503152 and EudraCT n. 2006-005954-62.

### Randomization and masking

After 1 month washout from any previous RAS blocking therapy (ACE inhibitor or ARB) and stratification by center, we randomly assigned included patients to treatment with benazepril, valsartan, or a combination of both medications on a 1:1:1 basis. Treatment was centrally allocated by an independent investigator (G.A.G.) who was not directly involved in conducting the study. The web-based, computer-generated randomization list was created using SAS (version 9.2). Blocking was used to ensure balance in the number of patients in each group at any time during the trial. The block sizes of 6 and 9 were randomly varied in order to increase the unpredictability of the sequence. Investigators involved in data handling and analyses were blinded to treatment allocation. A blinded-to-treatment adjudicating group reviewed the data to determine which patients had reached study endpoints and to evaluate safety.

### Procedures

At inclusion, we randomized patients to benazepril 10 mg/day, valsartan 160 mg/day or to benazepril 5 mg/day and valsartan 80 mg/day combination therapy. These doses correspond to half or one-fourth, respectively, of the full doses recommended by the manufacturer for BP control [15,16]. Moreover, 7 to 10 days later, we monitored BP, serum creatinine, serum potassium, blood glucose, and venous pH. If treatment was tolerated well, we up-titrated benazepril to 20 mg/day, valsartan to 320 mg/day, and benazepril/valsartan combination therapy to 10/160 mg/day. Furthermore, seven to 10 days after up-titration, we again monitored the aforementioned variables. If well tolerated, we continued treatment during the whole study period. In patients with adverse events possibly related to treatment, we back-titrated the dose of the study drugs to the previous step. Additional BP-lowering medications were allowed to achieve and maintain the target BP according to the following steps: (1) thiazide or loop diuretics; (2) clonidine or beta and/or alpha blockers; and (3) dihydropyridine calcium channel blockers, or minoxidil. In case of symptomatic hypotension or serum potassium $\geq$5.5 mEq/L in spite of diuretic therapy and optimized blood glucose and acid/base balance, back titration of concomitant treatments, and, secondarily, of the study drugs were allowed. Treatment goal was to maintain target BP with the highest dose of benazepril and/or valsartan and the lowest doses of the concomitant BP-lowering drugs. Potassium-sparing diuretics, aldosterone antagonists, and RAS inhibitors different from the study drugs were not allowed. Concomitant treatment was targeted to HbA1c <7% (<53 mmol/mol) and total cholesterol <200 mg/dL. Baseline parameters, including measured GFR [17] for patients referred to the CRC, were assessed at 3 and 6 months after randomization and every 6 months thereafter. At randomization, the GFR was also estimated by the Chronic Kidney Disease Epidemiology Collaboration (CKD-EPI) equation. Measurements of albuminuria by nephelometry (Immage; Beckman Coulter, Milano, Italy) and GFR by the iohexol plasma clearance technique [17] were centralized at the laboratories of the CRC.

### Outcome measures

The primary endpoint was the development of persistent microalbuminuria (defined as UAE rate 20 to 199 μg/min in at least 2 of 3 consecutive overnight urine collections confirmed in 2 consecutive visits) over a planned median follow-up of 4.5 years. Secondary outcomes

included albuminuria considered as a continuous variable and major cardiovascular events (MACEs), including sudden cardiovascular death and fatal and nonfatal acute myocardial infarction or stroke. Non-prespecified analyses considered in addition to MACE included coronary or peripheral revascularization or amputation because of critically ischemic limb. Safety variables included serious and nonserious adverse events and any clinical or laboratory abnormality possibly related to the study drugs.

## Sample size and statistical analysis

Primary outcome analysis aimed to compare the incidence of microalbuminuria between patients randomized to benazepril monotherapy and those randomized to benazepril plus valsartan combination therapy. Based on the outcome data of the BENEDICT [3] and on aggregate data of the VARIETY study evaluated in blinded fashion, we estimated that 27% of patients in the ACE inhibitor group would have developed persistent microalbuminuria over 4.5 years of observation period. Similar figures were expected for patients in the ARB group. Based upon preliminary evidence in type 2 diabetes [18], this figure was, conservatively, predicted to decrease by 45% (from 27% to 14.85%) with combination therapy as compared to benazepril or valsartan monotherapy. To give the trial an 80% power to detect as statistically significant (alpha = 0.05, 2-tailed test) the expected between-group difference in microalbuminuria incidence, and accounting for a 5% drop-out rate, 188 patients per group had to be included for a total of 564 patients. All statistical analyses were intention to treat by using SAS (version 9.4) and Stata (version 15). For the analysis of the primary efficacy endpoint, the full set of data was used, including data of all randomized patients, with the exception of 10 (6 assigned to benazepril and 4 to combination therapy) who were found to be microalbuminuric at the time of randomization and 2 additional patients both allocated to combination therapy without baseline UAE evaluation. For safety and secondary efficacy endpoints, data from all randomized patients were used.

Time to the primary endpoint (new onset of persistent microalbuminuria) was calculated with the interval-censored method. With this approach, the event is considered to occur between the last visit with documented normoalbuminuria to the first visit with evidence of validated microalbuminuria [3]. The primary analysis was performed by the accelerated failure time model, implemented with the SAS PROC LIFEREG procedure, which allowed the direct incorporation of interval-censored data, and by use of proportional hazards model, implemented with the SAS PROC PHREG. This approach allowed the use of interval-censored data that addresses the issue that the "exact date" of development of microalbuminuria was not known. The only information available was that the endpoint was reached within a time interval ranging from the last visit the participant was found to be normoalbuminuric to the first visit he/she was found to be microalbuminuric.

The following prespecified baseline covariates were included in the model: study center, patient age and sex, smoking habit (never versus current and former), baseline mean BP, and log-transformed UAE. The magnitude of the treatment effect was assessed by calculating the accelerator factor, which quantifies the effect of one treatment relative to another in accelerating or slowing disease progression, as previously described [3]. A Cox regression model was also applied to ensure the robustness of the results by using as the assumed date of development of microalbuminuria the midpoint of the interval between the last visit when the participant was normoalbuminuric and the first visit when he/she was microalbuminuric. Proportionality assumptions were assessed using Schoenfeld residuals. For graphical representation, Kaplan–Meier curves were plotted for each treatment group, using the midpoint of the interval as event times. The main treatment comparison was between combination therapy

versus benazepril monotherapy, and the 5% level of significance was used for this comparison. Secondary comparisons were between combined therapy and valsartan monotherapy and between benazepril monotherapy and valsartan monotherapy, and these comparisons were conducted at the 2.5% level of significance (2-sided test) to take into account multiple testing.

Non-prespecified, exploratory analyses were performed to assess treatment effects on the composite cardiovascular outcome of sudden cardiac death, fatal and nonfatal acute myocardial infarction and stroke plus coronary or peripheral revascularization, or amputation because of critically ischemic limb.

GFR slope (single-slope linear model) was calculated in patients who had at least 2 GFR measurements in addition to baseline value. Slopes were the regression lines between GFR measurements and time. Between-group differences in GFR slopes were assessed by Wilcoxon rank sum test.

One interim analysis was performed on the intention-to-treat population on May 2015 (see study protocol). Adverse events were classified using the Medical Dictionary for Regulatory Activities (version 21.0). Data were presented as number (%), mean (SD), or median [interquartile range, IQR], as appropriate. $P < 0.05$ (2-sided) was considered statistically significant.

## Results

Of the 1,061 patients assessed for eligibility, 449 were excluded. Overall, 408 patients—including 2 with a concomitant disease that could confound data interpretation—did not fulfill the selection criteria. In addition, 32 patients withdrew their informed consent, 8 were lost to follow-up, and 1 was excluded due to initiation of ARB therapy after the run-in period. The remaining 612 patients were enrolled between July 2007 and April 2013 (Fig 1). A total of 209 patients were randomly assigned to benazepril (10 mg/day), 202 to valsartan (80 mg/day) plus benazepril (5 mg/day) combination therapy, and 201 to valsartan (160 mg/day). Baseline demographic, clinical, and laboratory characteristics were similar among groups, as was the distribution of concomitant medications (Table 1). A total of 77 of the 86 patients who were referred to the CRC consented to serial GFR measurements. Their characteristics were similar among treatment groups (S1 Table).

Study treatments were up-titrated to full maintenance doses (benazepril 20 mg/day; valsartan 160 mg/day plus benazepril 10 mg/day; or valsartan 320 mg/day) in 205, 156, and 199 patients of those randomized to benazepril (98.1%), combination therapy (77.2%), and valsartan (99.0%), respectively. The last follow-up visit was completed on July 2016. Throughout the follow-up, 17 patients died, and 128 withdrew from the study because of adverse events ($n = 16$), consent withdrawal ($n = 83$), loss to follow-up ($n = 14$), poor compliance ($n = 3$), or other reasons ($n = 12$) (Fig 1).

### Main renal endpoints

During a median [IQR] follow-up of 66 [42 to 83] months, the primary endpoint of persistent microalbuminuria was reached by 57 (28.1%) patients on benazepril, 53 (27.0%) on combination therapy, and 64 (31.8%) on valsartan. The estimated acceleration factors were 1.410 (95% CI: 0.806 to 2.467, $P = 0.229$) for benazepril compared to combination therapy, 0.799 (95% CI: 0.422 to 1.514, $P = 0.492$) for benazepril compared to valsartan, and 1.665 (95% CI: 1.007 to 2.746, $P = 0.047$) for valsartan compared to combination therapy. After adjustment for predefined confounders, the estimated acceleration factors were 1.330 (95% CI: 0.784 to 2.255, $P = 0.290$) for benazepril compared to combination therapy, 1.051 (95% CI: 0.591 to 1.866, $P = 0.866$) for benazepril compared to valsartan, and 1.365 (95% CI: 0.873 to 2.132, $P = 0.172$) for valsartan compared to combination therapy. When using the Cox regression model, the

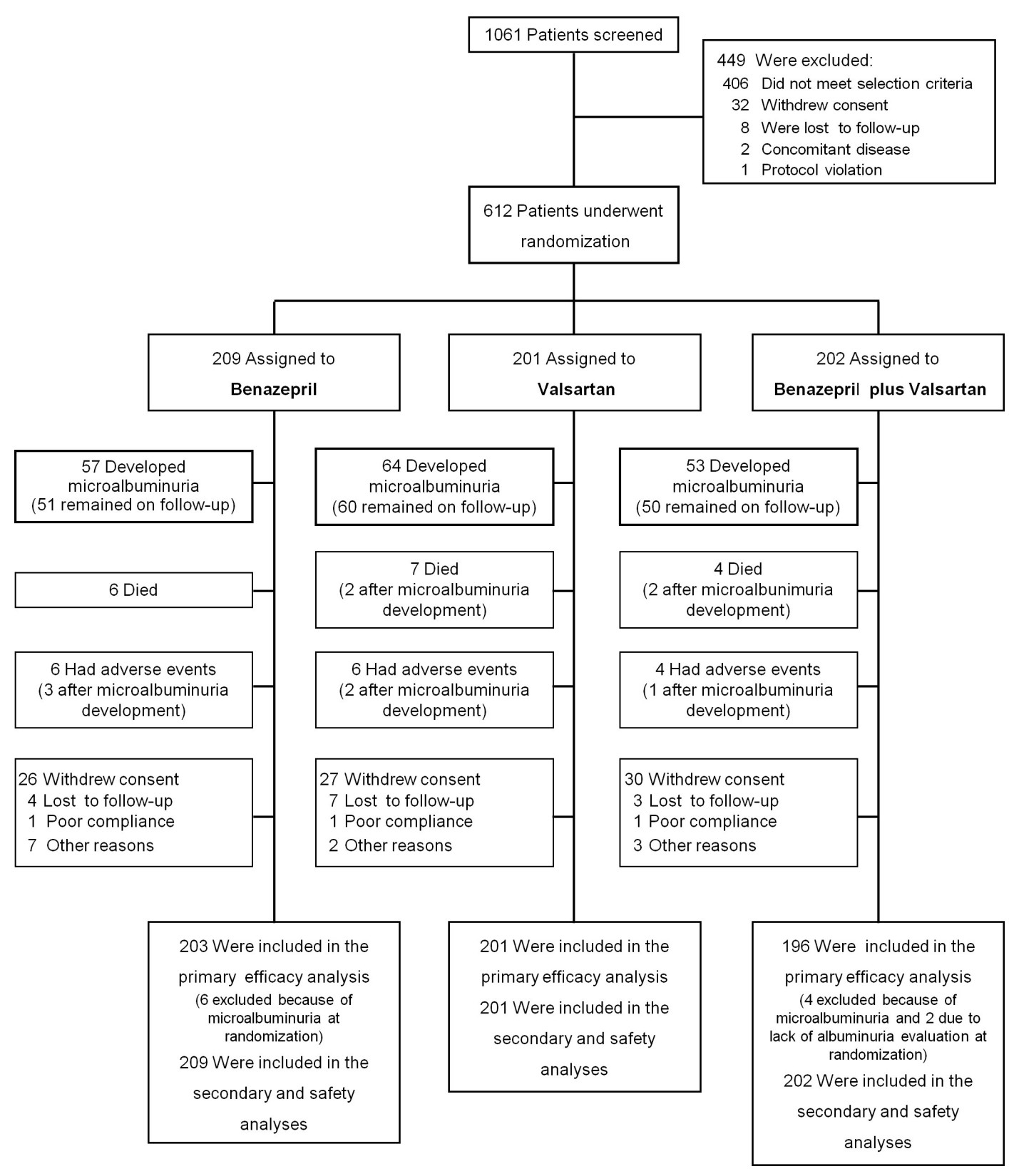

**Fig 1. Trial profile.**

**Table 1. Patient characteristics at baseline according to treatment group.**

| | Benazepril (*n* = 209) | Valsartan (*n* = 201) | Combination (*n* = 202) |
|---|---|---|---|
| **Demographic characteristics** | | | |
| Age, years | 64.7 ± 7.7 | 64.3 ± 7.9 | 65.0 ± 7.1 |
| Male sex, *n* (%) | 146 (69.9) | 138 (68.7) | 145 (71.8) |
| Known duration of diabetes, years | 11.5 ± 7.1 | 11.7 ± 6.9 | 12.2 ± 7.1 |
| Smoking status, *n* (%) | | | |
| Never smoked | 95 (45.4) | 102 (50.8) | 97 (48.0) |
| Former smoker | 34 (16.3) | 23 (11.4) | 32 (15.8) |
| Current smoker | 80 (38.3) | 76 (37.8) | 73 (36.2) |
| **Clinical features** | | | |
| BMI, kg/m$^2$ | 29.8 ± 4.4 | 30.3 ± 4.9 | 30.4 ± 4.9 |
| Systolic BP, mm Hg | 141.9 ± 14.6 | 140.9 ± 15.2 | 141.9 ± 13.7 |
| Diastolic BP, mm Hg | 78.7 ± 8.5 | 79.0 ± 7.3 | 78.8 ± 8.8 |
| MAP, mm Hg | 99.8 ± 8.8 | 99.7 ± 8.0 | 99.8 ± 8.6 |
| **Laboratory parameters** | | | |
| HbA1c, mmol/mol | 54.3 ± 14.2 | 54.3 ± 14.1 | 54.7 ± 12.9 |
| HbA1c, % | 7.1 ± 1.3 | 7.1 ± 1.3 | 7.2 ± 1.2 |
| Serum glucose, mg/dL | 151.4 ± 44.8 | 151.1 ± 50.6 | 152.5 ± 41.0 |
| Serum potassium, mg/dL | 4.13 ± 0.42 | 4.11 ± 0.48 | 4.10 ± 0.46 |
| Hemoglobin, g/dL | 14.0 ± 1.3 | 14.0 ± 1.2 | 14.1 ± 1.2 |
| Total cholesterol, mg/dL | 177.7 ± 33.0 | 178.4 ± 32.4 | 179.5 ± 34.6 |
| HDL cholesterol, mg/dL | 46.8 ± 12.2 | 48.0 ± 13.3 | 47.1 ± 11.5 |
| LDL cholesterol, mg/dL | 108.0 ± 28.2 | 108.4 ± 30.6 | 111.2 ± 32.3 |
| Triglycerides, mg/dL | 128.5 ± 72.1 | 127.9 ± 79.0 | 132.7 ± 76.4 |
| **Kidney function parameters** | | | |
| Serum creatinine, mg/dL | 0.90 ± 0.19 | 0.88 ± 0.19 | 0.93 ± 0.20 |
| Estimated GFR, mL/min/1.73 m$^{2\dagger}$ | 82.22 ± 15.28 | 84.19 ± 14.62 | 80.57 ± 15.99 |
| Measured GFR, mL/min/1.73 m$^{2*}$ | 85.93 ± 14.30 | 84.68 ± 20.85 | 83.04 ± 16.76 |
| UAE, μg/min | 8.74 [6.52 to 12.58] | 9.44 [6.72 to 12.59] | 8.35 [6.10 to 11.77] |
| **Patients with medications, *n* (%)** | | | |
| **- Antihypertensive agents** | | | |
| - Any | 168 (80.4) | 164 (81.6) | 158 (78.2) |
| - Diuretics | 106 (50.7) | 98 (48.7) | 91 (45.0) |
| - Calcium channel blockers | 96 (45.9) | 87 (43.3) | 81 (40.1) |
| - Beta-blockers | 51 (24.4) | 47 (23.4) | 51 (25.2) |
| - Sympatholytic agents | 0 | 0 | 0 |
| - ACE inhibitors | 1 (0.5) | 3 (1.5) | 1 (0.5) |
| - ARB | 2 (1.0) | 1 (0.5) | 0 |
| **- Lipid-lowering agents** | | | |
| - Any | 117 (56.0) | 110 (54.7) | 111 (54.9) |
| - Statins | 106 (50.7) | 98 (48.8) | 97 (48.0) |
| - Fibrates | 7 (3.3) | 12 (6.0) | 10 (5.0) |
| **- Hypoglycemic agents** | | | |
| - Any | 194 (92.8) | 192 (95.5) | 192 (95.0) |
| - Oral hypoglycemic agents | 175 (83.7) | 181 (90.0) | 175 (86.6) |
| - Insulin | 53 (25.4) | 44 (21.9) | 53 (26.2) |
| - Diet alone | 15 (7.2) | 9 (4.5) | 10 (5.0) |

Data are mean ± SD, median [IQR], or numbers (percentages).

$^\dagger$Estimated by the CKD-EPI equation.

$^*$Measured by iohexol plasma clearance in a subgroup of 77 patients.

Glycated hemoglobin (HbA1c) values were expressed by using percentage (%) units according to the DCCT and mmol/mol units according to the IFCC.

ACE, angiotensin converting enzyme; ARB, angiotensin receptor blocker; BMI, body mass index; BP, blood pressure; CKD-EPI, Chronic Kidney Disease Epidemiology Collaboration; DCCT, Diabetes Control and Complication Trial; GFR, glomerular filtration rate; HbA1c, glycated hemoglobin; HDL, high-density lipoprotein; IFCC, International Federation of Clinical Chemistry and Laboratory Medicine; IQR, interquartile range; LDL, low-density lipoprotein; MAP, mean arterial pressure; UAE, urinary albumin excretion.

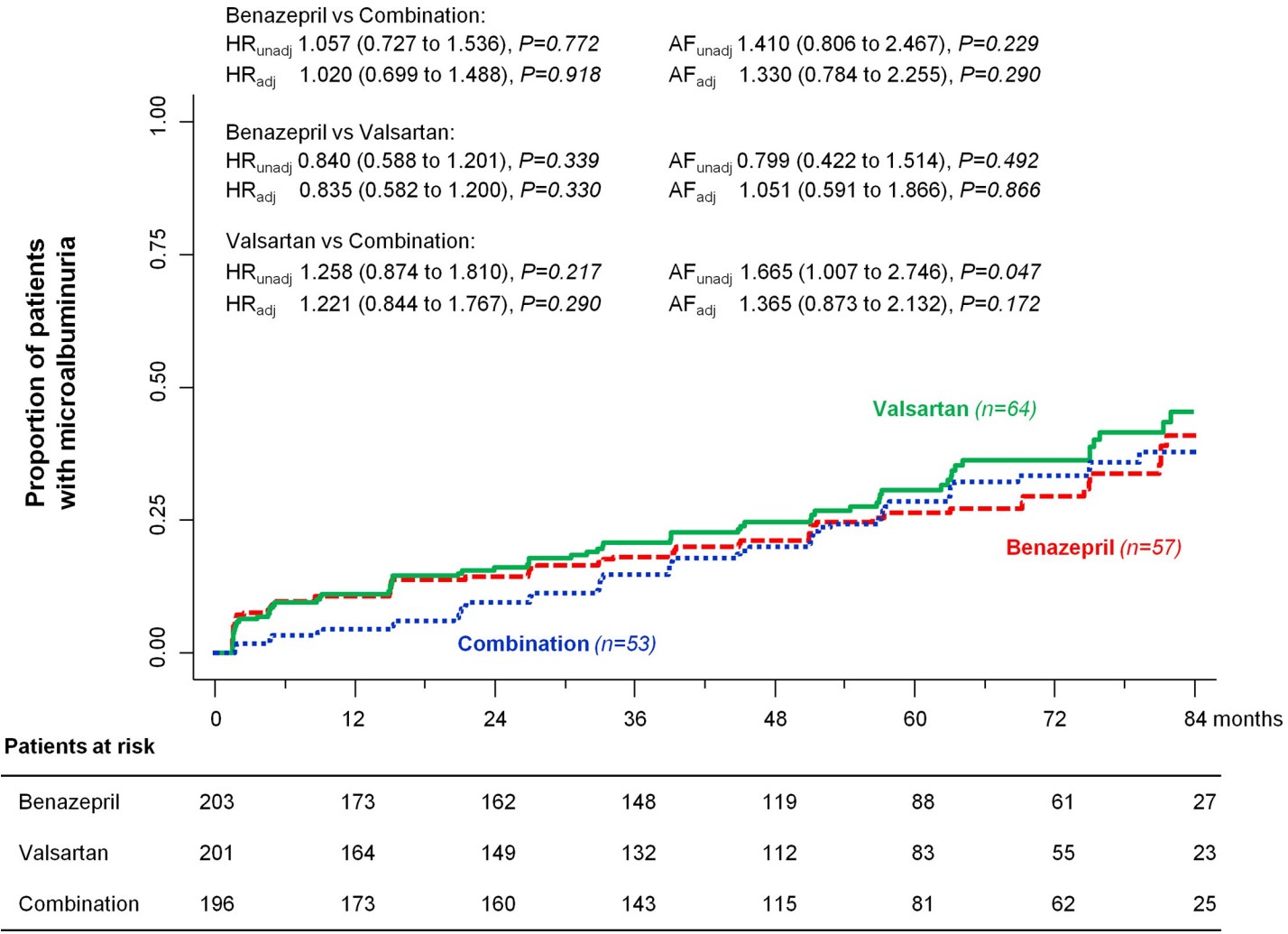

**Fig 2. Kaplan–Meier curves for the primary endpoint of progression to microalbuminuria.** Kaplan–Meier curves show the proportion of patients who reached the primary endpoint of progression to microalbuminuria in the benazepril, valsartan, and combination therapy groups during a median follow-up of 66 months. HRs, AFs, and their respective 95% confidence intervals are crude (unadjusted) and adjusted for center, age, sex, smoking habit, baseline mean BP, and log-transformed UAE. Adj, adjusted; AF, acceleration factor; BP, blood pressure; HR, hazard ratio; UAE, urinary albumin excretion; unadj, unadjusted.

hazard for new-onset microalbuminuria was similar with the 3 treatment regimens, even after adjustment for predefined features (Fig 2).

### Exploratory analyses for the renal endpoints

The incidence of microalbuminuria was similar in regimens that included benazepril alone or combined to valsartan, and in those not comprising the ACE inhibitor, also after adjusting for predefined covariates (S1A Fig). Consistent with this finding, the event rate was comparable between study treatments that did not include valsartan and in valsartan-based regimens, even following adjustment for predefined confounders (S1B Fig).

At univariable analysis, older age, male sex, and higher levels of log-transformed UAE, HbA1c, serum creatinine, and triglycerides at baseline were significantly associated with increased risk for microalbuminuria development. At multivariable analysis, older age, log-transformed UAE, serum levels of creatinine, and triglycerides were independently associated with subsequent development of microalbuminuria (S2 Table).

## Secondary endpoints

During the follow-up, median UAE did not differ significantly among groups at any time point (Fig 3A). In the subgroup of patients in whom GFR was measured by iohexol plasma clearance, GFR declined by 1.78 [0.32 to 3.54] mL/min/1.73 m$^2$ per year. The annual rate of GFR decline did not differ significantly among the benazepril (1.73 [0.44 to 4.13] mL/min/1.73 m$^2$), combination therapy (2.61 [0.77 to 3.59] mL/min/1.73 m$^2$), or valsartan (1.45 [−0.94 to 2.92] mL/min/1.73 m$^2$) groups.

During the study period, 7 (3.3%) patients on benazepril, 9 (4.5%) on combination therapy, and 9 (4.5%) on valsartan reached the composite cardiovascular endpoint of sudden cardiac death and fatal and nonfatal acute myocardial infarction or stroke. The risk of progression to the combined endpoint was similar between groups, even after adjusting for predefined confounders (Fig 4).

## Exploratory analyses for the secondary endpoints

During the observation period, 13 (6.2%) patients on benazepril, 17 (8.4%) on combination therapy, and 16 (8.0%) on valsartan progressed to the explorative composite cardiovascular endpoint of sudden cardiac death, fatal or nonfatal acute myocardial infarction and stroke in addition to coronary or peripheral revascularization, or amputation because of critically ischemic limb. The event rate was similar among treatment groups, even after adjusting for predefined covariates (S2 Fig).

## BP and glycemic control

BP was remarkably similar across groups at baseline and throughout the study period (Fig 3B). During the follow-up, the mean differences in systolic BP were 0.16 mm Hg (95% CI: −1.93 to 2.25, $P = 0.881$) for benazepril versus combination therapy, 0.97 mm Hg (95% CI: −1.11 to 3.06, $P = 0.360$) for benazepril versus valsartan, and −0.81 mm Hg (95% CI: −3.00 to 1.38, $P = 0.467$) for valsartan versus combination therapy. As for diastolic BP, mean differences were 0.86 mm Hg (95% CI: −0.43 to 2.15, $P = 0.190$) for benazepril versus combination therapy, 0.82 mm Hg (95% CI: −0.43 to 2.07, $P = 0.199$) for benazepril versus valsartan, and 0.04 mm Hg (95% CI: −1.29 to 1.38, $P = 0.949$) for valsartan versus combination therapy. During the study period, mean differences in HbA1c levels were −0.06 mmol/mol (95% CI: −2.35 to 2.23, $P = 0.957$) for benazepril versus combination therapy, −1.46 mmol/mol (95% CI: −3.81 to 0.89, $P = 0.224$) for benazepril versus valsartan, and 1.39 mmol/mol (95% CI: −0.84 to 3.62, $P = 0.221$) for valsartan versus combination therapy (Fig 3C). Changes of HbA1c in the valsartan group tended to mirror the concomitant changes in Hb concentration (S3 Fig).

## Safety

A total of 63 (30.1%) patients on benazepril, 52 (25.9%) on valsartan, and 56 (27.7%) on combination therapy had at least 1 serious adverse event. Overall, the distribution of serious (Table 2) and nonserious (S3 Table) adverse events was similar in the 3 treatment groups.

During the follow-up, 14 (6.7%) patients on benazepril, 16 (8.0%) on valsartan, and 14 (6.9%) on combination therapy permanently discontinued the study treatment. Two (1.0%) patients on benazepril, 4 (2.0%) on valsartan, and 13 on combination therapy (6.4%, $P = 0.003$ versus benazepril, $P = 0.044$ versus valsartan) experienced 1 or more episodes of treatment-related hyperkalemia. Only 1 of these events was serious and led to treatment withdrawal in a patient on benazepril. Treatment-related hypotension was reported at least once by 9 (4.3%) patients on benazepril, 15 (7.5%) on valsartan, and 20 on combination therapy (9.9%,

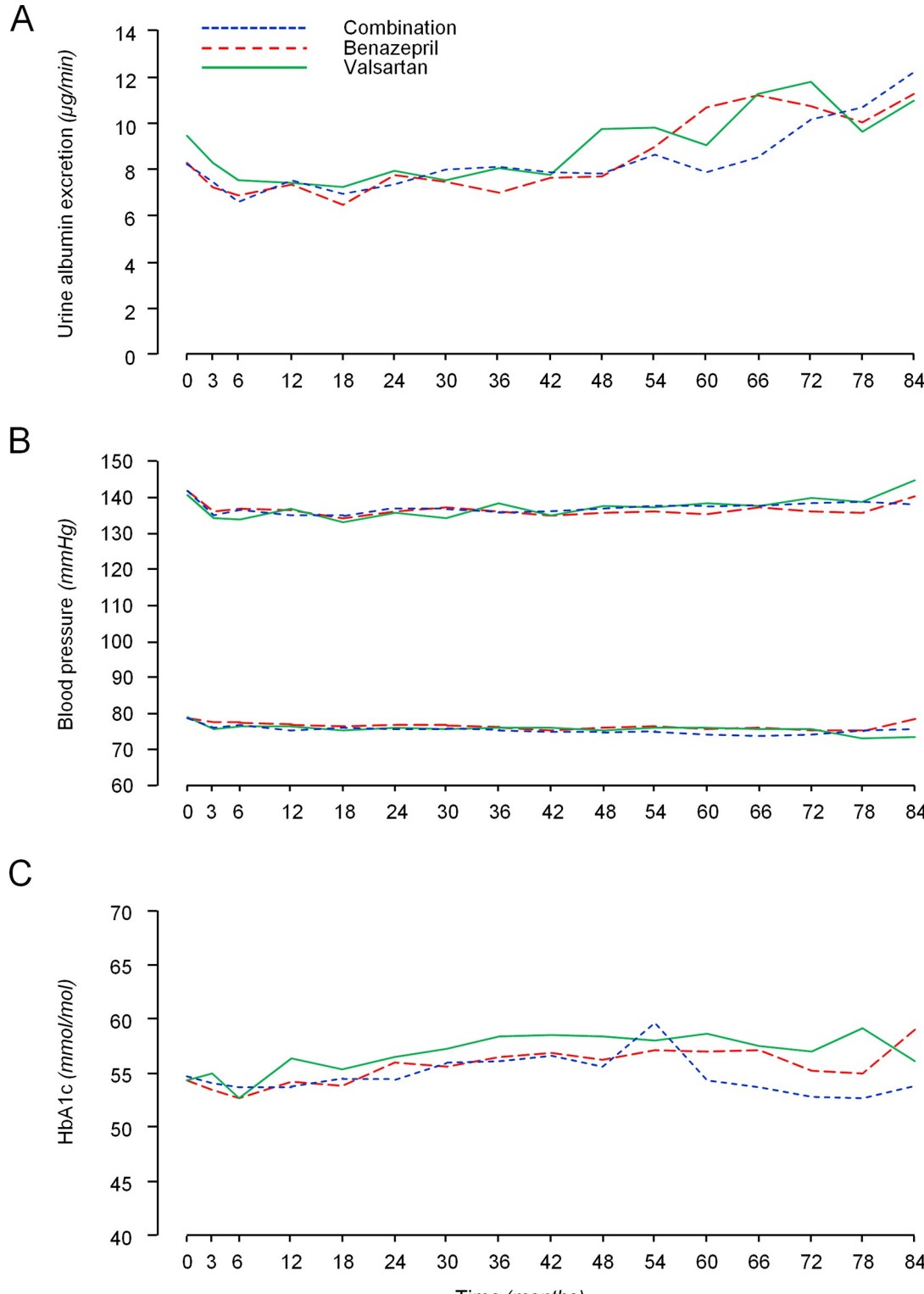

**Fig 3. Median albuminuria, mean BP, and HbA1c levels during the study period according to treatment groups.** Median albuminuria (**A**), mean systolic and diastolic BP (**B**), and HbA1c levels (**C**) during the study period according to treatment groups. BP, blood pressure.

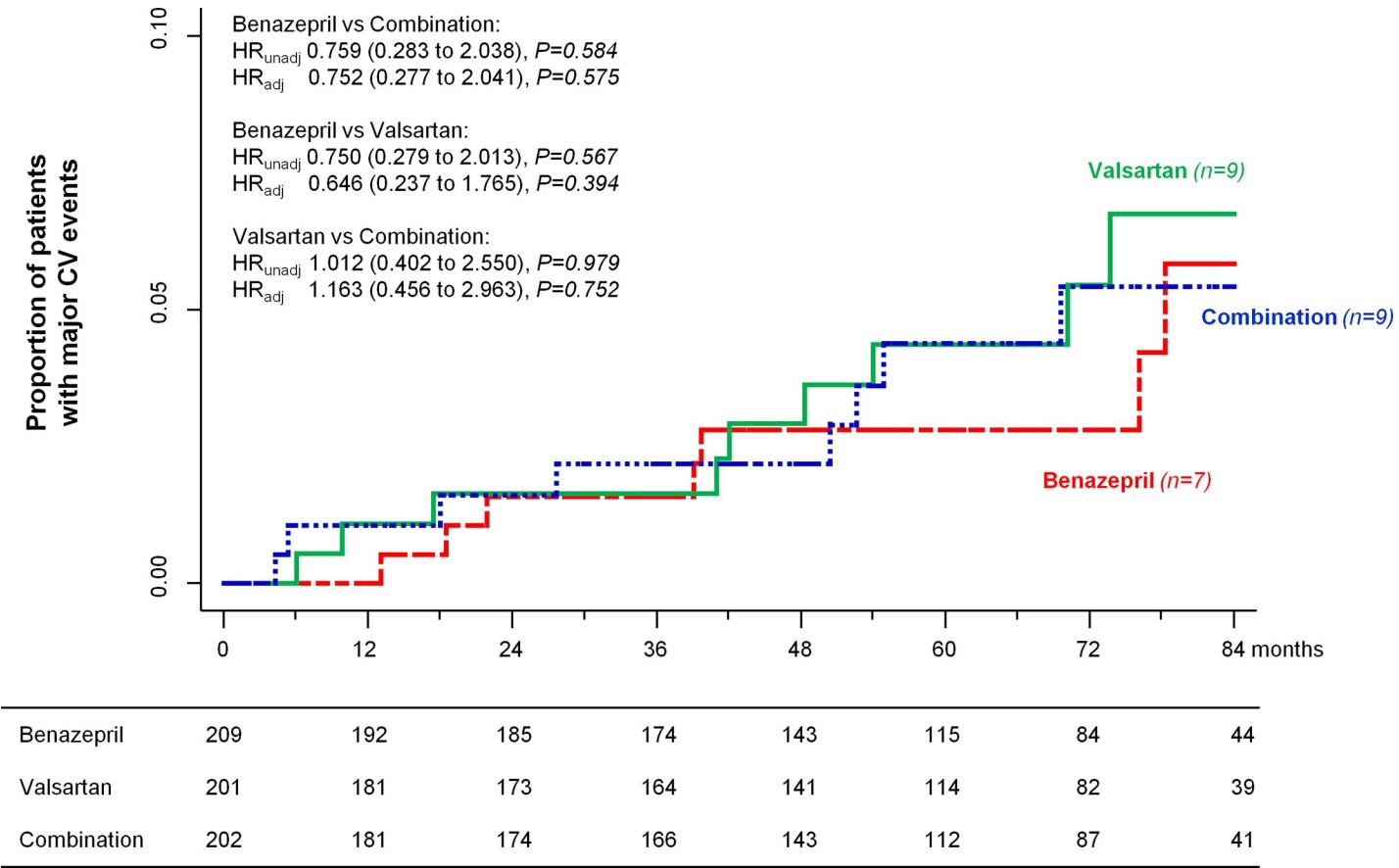

**Fig 4. Kaplan–Meier curves for fatal or nonfatal MACEs.** Kaplan–Meier curves show the proportion of patients who reached the composite endpoint of sudden cardiac death and fatal and nonfatal acute myocardial infarction or stroke in the benazepril, valsartan, and combination therapy groups during a median follow-up of 66 months. HRs and 95% confidence intervals are crude (unadjusted) and adjusted for center, age, sex, smoking habit, baseline mean BP, and log-transformed UAE. Adj, adjusted; BP, blood pressure; HR, hazard ratio; MACE, major cardiovascular event; UAE, urinary albumin excretion; unadj, unadjusted.

$P$ = 0.033 versus benazepril). None of these cases was serious, but in 2 patients in the valsartan group, treatment had to be withdrawn. Five patients on benazepril (2.4%), 1 on valsartan (0.5%), and 5 on combination therapy (2.5%) experienced treatment-related cough. These events were neither serious nor mandated treatment discontinuation. There were 10 deaths not attributed to cardiovascular events: 4 on benazepril, 5 on valsartan, and 1 on combination therapy (Table 2). These events were deemed not related to the study treatments.

## Discussion

In this PROBE trial, we found that, at comparable BP control, combination therapy with half of the standard manufacturer-recommended antihypertensive doses of benazepril and valsartan had no superior protective effect against progression to microalbuminuria as compared to monotherapy with full recommended doses of either benazepril or valsartan in a homogeneous cohort of type 2 diabetic patients at high risk of progression to the primary endpoint because of hypertension and high-normal albuminuria (UAE ≥7 and <20 μg/min) to start with. The protective effect against progression to microalbuminuria of benazepril and valsartan monotherapy was also comparable. Also, changes in albuminuria, considered as a continuous variable, were similar across groups. Consistently, in the subgroup of patients with available serial

**Table 2. Number of patients with at least 1 SAE according to treatment group.**

| Patients with SAEs, *n* | Benazepril (*n* = 209) | Valsartan (*n* = 201) | Combination (*n* = 202) |
|---|---|---|---|
| **Fatal** | | | |
| MACEs* | | | |
| -Myocardial infarction | 1 | 1 | 0 |
| -Stroke | 0 | 1 | 0 |
| -Sudden cardiac death | 1 | 0 | 3 |
| Neoplasia‡ | 2 | 4 | 1 |
| Traffic accident | 1 | 0 | 0 |
| Suicide | 0 | 1 | 0 |
| Respiratory failure | 1 | 0 | 0 |
| **Nonfatal** | | | |
| **Cardiovascular** | | | |
| *MACEs** | | | |
| -Myocardial infarction | 2 | 3 | 4 |
| -Stroke | 1 | 3 | 2 |
| *Minor cardiovascular events†* | | | |
| -Transitory ischemic attack | 0 | 0 | 1 |
| *Other cardiovascular events* | | | |
| -Peripheral artery disease/revascularization | 2 | 4 | 5 |
| -Peripheral artery disease/amputation | 1 | 0 | 1 |
| -Coronary revascularization | 6 | 7 | 6 |
| -Atrial fibrillation | 3 | 1 | 0 |
| -Atrial flutter | 1 | 1 | 0 |
| -Other electrocardiographic anomalies | 3 | 0 | 3 |
| -Syncope | 1 | 2 | 1 |
| -Heart failure | 1 | 3 | 2 |
| -Unstable angina | 1 | 0 | 1 |
| -Aortic valve insufficiency | 0 | 0 | 1 |
| -Pericardial effusion | 0 | 0 | 1 |
| -Peripheral venous insufficiency | 0 | 1 | 1 |
| -Pulmonary embolism | 0 | 1 | 0 |
| **Renal** | | | |
| -Acute kidney injury | 4 | 0 | 2 |
| -Chronic kidney disease | 2 | 0 | 0 |
| -Nephrotic syndrome | 0 | 0 | 1 |
| **Urological** | | | |
| -Urolithiasis | 5 | 2 | 1 |
| -Benign prostatic hyperplasia | 3 | 2 | 1 |
| -Prostatitis | 0 | 1 | 0 |
| -Urethritis | 1 | 0 | 0 |
| -Other urological events | 5 | 2 | 0 |
| **Cancer and benign tumors** | | | |
| -Prostate cancer | 2 | 2 | 1 |
| -Skin cancer | 2 | 1 | 1 |
| -Colorectal cancer | 1 | 2 | 0 |
| -Pancreatic cancer | 2 | 0 | 0 |
| -Bladder cancer | 1 | 0 | 0 |
| -Hepatocellular carcinoma | 0 | 1 | 0 |

(*Continued*)

**Table 2.** (Continued)

| Patients with SAEs, n | Benazepril (n = 209) | Valsartan (n = 201) | Combination (n = 202) |
|---|---|---|---|
| -Breast cancer | 0 | 1 | 0 |
| -Gastric cancer | 0 | 0 | 1 |
| -Non-Hodgkin lymphoma | 0 | 0 | 1 |
| -Keratoacanthoma | 1 | 0 | 0 |
| -Colon adenoma | 1 | 0 | 0 |
| -Schwannoma | 1 | 0 | 0 |
| -Lung cancer | 0 | 2 | 0 |
| **Gastrointestinal** | | | |
| -Cholelithiasis/cholecystitis | 2 | 1 | 5 |
| -Acute pancreatitis | 1 | 0 | 1 |
| -Acute diverticulitis | 2 | 1 | 2 |
| -Intestinal obstruction | 0 | 2 | 1 |
| -Diarrhea | 0 | 1 | 1 |
| -Other gastrointestinal events | 1 | 1 | 1 |
| **Respiratory** | | | |
| -Pneumonia | 0 | 1 | 1 |
| -Acute bronchitis | 0 | 0 | 1 |
| -Sleep apnea syndrome | 1 | 0 | 0 |
| -Dyspnea | 0 | 1 | 0 |
| **Neurological** | | | |
| -Vertigo | 1 | 1 | 1 |
| -Epilepsy | 0 | 0 | 1 |
| -Subdural hematoma | 0 | 0 | 2 |
| -Other neurological events | 0 | 2 | 3 |
| **Metabolic** | | | |
| -Decompensated diabetes | 2 | 0 | 3 |
| -Hyperkalemia | 1 | 0 | 0 |
| -Anemia | 0 | 1 | 2 |
| -Other metabolic events | 2 | 0 | 1 |
| **Musculoskeletal and trauma** | | | |
| -Musculoskeletal pain | 0 | 1 | 2 |
| -Repetitive muscular strain injury | 1 | 1 | 3 |
| -Osteoarthritis | 4 | 5 | 6 |
| -Intravertebral disc hernia | 0 | 0 | 3 |
| -Abdominal hernia | 1 | 1 | 1 |
| -Bone fracture | 3 | 1 | 3 |
| -Traumatic amputation | 0 | 0 | 1 |
| -Head injury | 2 | 0 | 2 |
| **Gynecological** | | | |
| -Benign uterine or adnexal mass | 1 | 0 | 1 |
| -Vaginal prolapse | 1 | 0 | 0 |
| **Infections** | | | |
| -Sepsis | 1 | 1 | 0 |
| -Herpes zoster | 0 | 1 | 0 |
| -Staphylococcal meningitis | 1 | 0 | 0 |
| -Subhepatic abscess | 0 | 0 | 1 |
| -Appendicitis | 0 | 0 | 1 |

(*Continued*)

**Table 2.** (Continued)

| Patients with SAEs, *n* | Benazepril (*n* = 209) | Valsartan (*n* = 201) | Combination (*n* = 202) |
|---|---|---|---|
| -Knee prosthetic joint infection | 1 | 0 | 0 |
| -Foot gangrene | 0 | 0 | 1 |
| -Amputation | 0 | 0 | 1 |
| **Other SAEs** | 1 | 1 | 2 |

No statistically significant difference observed across treatments.

[*]**Major CV events:** Sudden cardiac death, fatal and nonfatal acute myocardial infarction, or stroke.

[†]**Minor CV events:** Transient ischemic attack and coronary artery disease without revascularization.

[‡]**Neoplasia:** Hodgkin lymphoma (*n* = 1), lung cancer (*n* = 2), colorectal cancer (*n* = 1), pancreatic cancer (*n* = 1), malignant neoplasm of retroperitoneum (*n* = 1), and metastases to the central nervous system of colorectal cancer (*n* = 1).

CV, cardiovascular; MACE, major cardiovascular event; SAE, serious adverse event.

GFR measurements with the iohexol plasma clearance technique [17], GFR decline was similar across treatment arms. Baseline demographic, clinical, and laboratory characteristics of the 3 study groups were quite similar. No systematic changes in diet or concomitant medications that could affect UAE were introduced throughout the study. Moreover, study findings were confirmed by multivariable analyses adjusted for predefined potential confounding factors, such as study center, patient age and sex, smoking habit, and baseline BP and UAE. BP and metabolic control were also similar in the 3 treatment groups throughout the whole study period. Indeed, the slight increases in HbA1c levels observed in the valsartan group most likely reflected the concomitant slight decreases in hemoglobin concentration observed in this group [19]. None of these changes was clinically relevant.

Consistently with the relatively low level of albuminuria at inclusion, good control of cardiovascular risk factors, in particular hypertension and diabetes, and, probably even more relevant, generalized RAS inhibitor therapy, the overall incidence of fatal and nonfatal cardiovascular events was low. The rate of cardiovascular events was also similar across groups. Conceivably, also this finding might reflect optimized cardioprotective treatment in all the 3 study groups. Admittedly, however, the number of patients and events were too small to draw any firm conclusion in this regard.

All treatments were safe and well tolerated. Distribution of serious and nonserious adverse events in the 3 groups was similar, with a slight excess of treatment-related episodes of hypotension or hyperkalemia in the combination therapy group. Cough was less frequently reported with valsartan monotherapy that, however, was associated with a trend to more episodes of symptomatic hypotension. Treatment-related adverse events were uncommon and were rarely serious or required treatment withdrawal in any considered treatment group.

The rationale of dual RAS blockade rests on the hypothesis that simultaneous inhibition of the converting enzyme and blockade of the angiotensin type 2 receptor could prevent both the production and the action of angiotensin II. Blockade of the cascade at 2 different levels (upstream and downstream angiotensin II) could achieve a synergistic inhibitory effect of the RAS resulting in a more effective reduction of albuminuria with combination therapy than with ACE inhibitor or ARB monotherapy. Moreover, combination therapy was expected to prevent the so-called "escape" of angiotensin II and aldosterone from RAS inhibition, a phenomenon resulting in recovery of angiotensin II and aldosterone plasma levels to baseline values, with consequent progressive exhaustion of renoprotective effects [20,21]. Indeed, several, small-scale studies with relatively short follow-up found that dual RAS blockade combining an ACE inhibitor and an ARB may reduce albuminuria more effectively than single blockade

with ACE inhibitor or ARB monotherapy in subjects with or without diabetes [9,22–24]. Conversely, the Candesartan and Lisinopril Microalbuminuria II (CALM II) trial showed that 12-month treatment with lisinopril 40 mg/day or candesartan 16 mg/day plus lisinopril 20 mg/day had comparable effects on UAE in hypertensive patients with diabetes and normo- or microalbuminuria at baseline [25]. Independent of the above inconsistencies, results of these studies were largely flawed by larger BP reduction achieved with combination therapy than with monotherapy [9,22–24]. Thus, whether the superior antialbuminuric effect of combination therapy was mediated by more effective inhibition of the RAS or rather by larger BP decline per se remained elusive. This is an issue with major practical implications because, in the case of a BP-driven effect, any combined therapy even with medications that do not directly interfere with the RAS would have stronger antiproteinuric effects than ACE inhibitor or ARB monotherapy. Thus, our present study provides the first evidence that ACE inhibitor plus ARB dual therapy and either agent alone at equivalent antihypertensive doses are similarly effective in the primary prevention of microalbuminuria in patients with type 2 diabetes and hypertension. Data from the LIRICO [26], VALID [5], and PRONEDI [12] trials suggest that these findings can be extended also to subjects with overt nephropathy.

## Limitations and strengths

The lower-than-expected albuminuria in our study population at inclusion unavoidably reduced the power of the analyses to detect a treatment effect on the primary endpoint. On the other hand, our finding that the incidence of microalbuminuria over 66-month follow-up was low and virtually identical across study groups suggests that, very unlikely, any difference among groups would have emerged with higher values of albuminuria at baseline. The conclusion that the 3 treatments were similarly renoprotective is further corroborated by evidence that all other outcomes assessed to evaluate renal disease progression, including albuminuria as a continuous variable and GFR, showed similar changes among treatments throughout the whole observation period. The lack of a control group without ACE inhibitor and ARB was justified by results of previous randomized controlled trials demonstrating a protective effect of either ACE inhibitor or ARB treatment over placebo against microalbuminuria development in patients with type 2 diabetes [27].

Major strengths were the direct head-to-head comparison of ACE inhibitor, ARB and their combination at similar BP control, the centralized UAE measurement by a gold standard procedure in triplicate overnight urine collections, and the long duration of the follow-up. Study findings have large external validity, since outcome data were obtained in patients who account for at least half of the whole diabetic population [28]. The robustness of the study findings was confirmed by the consistency of results of both the accelerated failure time model and the Cox proportional hazards model.

In conclusion, treatment with 10 mg/day of benazepril plus 160 mg/day of valsartan did not slow progression to microalbuminuria as compared to either 20 mg/day of benazepril or 320 mg/day of valsartan in hypertensive patients with type 2 diabetic and high-normal UAE. Also, the safety profile of the 3 tested regimens was quite similar with a slight excess, if any, of side effects with combination therapy. Thus, dual RAS blockade should not be preferred to ACE inhibitor or ARB monotherapy for the prevention of microalbuminuria in normoalbuminuric type 2 diabetics. Recent studies showing that drugs that inhibit the sodium–glucose co-transport at proximal tubular levels (sodium–glucose co-transporter 2 [SGLT2] inhibitors) may offer substantial nephro- and cardioprotection to patients with type 2 diabetes and different levels of albuminuria [29,30] might pave the way to novel prevention strategies based on the

integrated use of these novel medications with an ACE inhibitor or an ARB, but not with their combination.

## Supporting information

**S1 Table. Baseline characteristics of patients with GFR measurement by iohexol plasma clearance according to treatment group.** GFR, glomerular filtration rate. (DOCX)

**S2 Table. Baseline predictors of microalbuminuria development at univariable and multivariable analyses.** (DOCX)

**S3 Table. Number (%) of patients with at least 1 non-SAE according to treatment arm by MedDRA SOC.** non-SAE, nonserious adverse event; SOC, system organ classification. (DOCX)

**S1 Fig. Kaplan–Meier curves for the primary endpoint of progression to microalbuminuria according to treatment with or without ACE inhibitor (A) and with or without ARB (B).** ACE, angiotensin converting enzyme; ARB, angiotensin receptor blocker. (TIF)

**S2 Fig. Kaplan–Meier curves for the non-prespecified explorative composite cardiovascular endpoint of sudden cardiac death and fatal and nonfatal acute myocardial infarction or stroke in addition to coronary or peripheral revascularization, or amputation because of critically ischemic limb.** Kaplan–Meier curves show the proportion of patients who reached the exploratory composite cardiovascular endpoint in the benazepril, valsartan, and combination therapy groups during a median follow-up of 66 months. HRs and 95% confidence intervals are crude (unadjusted) and adjusted for center, age, sex, smoking habit, baseline mean BP, and log-transformed UAE. Adj, adjusted; BP, blood pressure; HR, hazard ratio; UAE, urinary albumin excretion; unadj, unadjusted. (TIF)

**S3 Fig. Mean hemoglobin levels during the study period according to treatment group.** (TIF)

**S1 Appendix. Study protocol.** (PDF)

**S2 Appendix. VARIETY Study Organization.** (DOCX)

**S3 Appendix. CONSORT checklist.** CONSORT, Consolidated Statement of Reporting Trials. (DOC)

**S4 Appendix. Ethics approval document.** (PDF)

## Acknowledgments

We thank the patients in the VARIETY study for their participation and contribution; the trial investigators, the diabetologists, nephrologists, and nurses of the participating centers for their invaluable assistance; and the laboratory and regulatory affairs staff, trial monitors, data managers and statisticians, and everyone at the Clinical Research Center for Rare Diseases Aldo e

Cele Daccò of the Istituto di Ricerche Farmacologiche Mario Negri IRCCS for their efforts in making this study possible. We thank Giuseppe Maiocchi and Federico Bertocchi (Novartis Farma, Origgio, Varese, Italy) for continuous support for the study and a major contribution to all the administrative and operational aspects concerning the supply and distribution of the study drugs.

## Author Contributions

**Conceptualization:** Piero Ruggenenti, Giuseppe Remuzzi.

**Data curation:** Piero Ruggenenti, Monica Cortinovis, Nadia Rubis.

**Formal analysis:** Annalisa Perna, Tobia Peracchi.

**Investigation:** Aneliya Parvanova, Matias Trillini, Ilian P. Iliev, Antonio C. Bossi, Antonio Belviso, Maria C. Aparicio, Roberto Trevisan, Stefano Rota, Silvia Prandini, Flavio Gaspari, Fabiola Carrara, Salvatore De Cosmo, Giancarlo Tonolo, Ruggero Mangili.

**Software:** Davide Martinetti.

**Supervision:** Piero Ruggenenti, Giuseppe Remuzzi.

**Writing – original draft:** Piero Ruggenenti, Monica Cortinovis, Giuseppe Remuzzi.

**Writing – review & editing:** Piero Ruggenenti, Monica Cortinovis, Aneliya Parvanova, Matias Trillini, Ilian P. Iliev, Antonio C. Bossi, Antonio Belviso, Maria C. Aparicio, Roberto Trevisan, Stefano Rota, Annalisa Perna, Tobia Peracchi, Nadia Rubis, Davide Martinetti, Silvia Prandini, Flavio Gaspari, Fabiola Carrara, Salvatore De Cosmo, Giancarlo Tonolo, Ruggero Mangili, Giuseppe Remuzzi.

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
