## [Editor Report · Decision Letter 0]

9 Apr 2021

Dear Dr Cortinovis, 

Thank you for submitting your manuscript entitled "Preventing microalbuminuria with benazepril, valsartan and benazepril-valsartan combination therapy in diabetic patients with high-normal albuminuria: a prospective, randomised, open label, blinded endpoint (PROBE) study" for consideration by PLOS Medicine.

Your manuscript has now been evaluated by the PLOS Medicine editorial staff and I am writing to let you know that we would like to send your submission out for external assessment.

However, before we can send your manuscript for assessment, we need you to complete your submission by providing the metadata that is required for full assessment. To this end, please login to Editorial Manager where you will find the paper in the 'Submissions Needing Revisions' folder on your homepage. Please click 'Revise Submission' from the Action Links and complete all additional questions in the submission questionnaire.

Please re-submit your manuscript within two working days, i.e. by Apr 13 2021 11:59PM.

Once your full submission is complete, your paper will undergo a series of checks in preparation for external assessment. 

Kind regards,

Richard Turner, PhD

rturner@plos.org

---

## [Decision Letter · Decision Letter 1]

28 Apr 2021

Dear Dr. Cortinovis,

Thank you very much for submitting your manuscript "Preventing microalbuminuria with benazepril, valsartan and benazepril-valsartan combination therapy in diabetic patients with high-normal albuminuria: a prospective, randomised, open label, blinded endpoint (PROBE) study" (PMEDICINE-D-21-01663R1) for consideration at PLOS Medicine. 

Your paper was discussed among the editorial team and sent to independent reviewers, including a statistical reviewer. The reviews are appended at the bottom of this email and any accompanying reviewer attachments can be seen via the link below:

[LINK]

In light of these reviews, we will not be able to accept the manuscript for publication in the journal in its current form, but we would like to invite you to submit a revised version that addresses the reviewers' and editors' comments fully. You will appreciate that we cannot make a decision about publication until we have seen the revised manuscript and your response, and we expect to seek re-review by one or more of the reviewers. 

We hope to receive your revised manuscript by May 19 2021 11:59PM. Please email us (plosmedicine@plos.org) if you have any questions or concerns.

Please let me know if you have any questions, and we look forward to receiving your revised manuscript. 

Sincerely,

Richard Turner, PhD

rturner@plos.org

Please add "GR is a member of PLOS Medicine's Editorial Board" or similar to your competing interest statement (article metadata). 

Please ensure that the data contact quoted is not an author of the ms (https://journals.plos.org/plosmedicine/s/data-availability). 

Please remove "methodologically sound" from your data statement. 

Please revisit the text at line 52 (abstract) - is the primary outcome not time to progression to microalbuminuria(page 11 of protocol)? As one or more of our referees comment, a few words of explanation about the meaning of acceleration factors would be helpful. 

Relatedly, we notice that HRs are quoted in Fig 2. Are HRs or acceleration factors the most relevant measure for the primary endpoint?

Please quote summary demographic details for study participants in your abstract.

After the abstract, please add a new and accessible "Author summary" section in non-identical prose. You may find it helpful to consult one or two recent research papers published in PLOS Medicine to get a sense of the preferred style. 

At line 289 and any other instances, please substitute "sex" for "gender" where appropriate.

Throughout the text please use the style "6 assigned to benzapril", although numbers should be spelt out at the start of sentences. 

Please remove the information on funding, competing interests and data sharing from the end of the main text. In the event of publication, this information will appear in the article metadata, via entries in the submission form. 

Please break the CONSORT checklist out into a separate supplementary file, labelled "S1_CONSORT_Checklist" or similar and referred to as such in the Methods section. 

In the checklist, please refer to individual items by section (e.g., "Methods") and paragraph number rather than by line or page numbers, as the latter generally change in the event of publication. 

Noting the statement about confidentiality in the attached protocol, we suggest substituting a version with the confidential information removed with your revision. 

Comments from the reviewers:

*** Reviewer #1: 

Congratulations to the authors and investigators for a well-conducted and clearly presented randomized trial in an area that still is fully elucidated, making the research question important. 

Comprehensive manuscript that leaves very few questions or comments:

Suggest explaining the concept of acceleration factors - this is an unfamiliar term. 

Suggest presenting decline in eGFR as annual, not monthly values, for a more clinical comparison. 

Table 1 Lab Parameters (and table S1): The HbA1c values are incorrect, as 5.6% does not correspond to 54 mmol/mol. Suggest to use http://www.ngsp.org/convert1.asp to convert between units

Page line 172: I believe the correct term is "glycated hemoglobin"

*** Reviewer #2: 

Statistical review

This paper reports a three-arm RCT comparing drugs for preventing microalbuminuria. The trial was conducted with a good length of follow-up using a robust design. The trial was generally reported well although there were some minor differences to the protocol/registration that should be noted - I have listed comments below.

1. Abstract "The estimated acceleration factors" - this might be clearer if the analysis model being a AFT model is mentioned

2. Abstract "BP control was similar across groups" - it's not clear why this is reported in the abstract - BP control is not listed as a trial outcome in the ClinicalTrials.gov registration.

3. Randomization and masking - I note the protocol mentions randomisation was stratified by centre, which is not stated here. Also adding the possible block sizes that were randomly chosen would be useful. 

4. Outcome measures: The cardiovascular events outcome was not listed on the registration page or in the protocol from what I can see. I would recommend it's made clear that this was no a pre-specified outcome. 

5. Sample size and statistical analysis: the authors mention 'based on the interim analysis' - can more information be provided about this? Was the sample size formally reassessed in this trial, and if so was it based on blinded or unblinded information?

6. Sample size and statistical analysis - I did not see where the valsartan monotherapy arm came into the analysis - the first line of this section implies this wasn't a primary analysis. If it was, the authors could comment on multiple testing, which is mentioned in the protocol but not in the paper.

7. Line 221 - I did not follow how the GFR slope was modelled - was difference between arms estimated? If so it is not reported in the results (just the estimated slope in each arm separately).

8. BP and glycemic control - please report these results with estimated differences between arms and CIs rather than just p-values.

James Wason

*** Reviewer #3: 

This manuscript reports the results of a well-designed and rigorously conducted RCT of benazepril vs. valsartan vs. combination therapy (at half dose) in participants with type 2 diabetes to evaluate the impact on the incidence of microalbuminuria. The manuscript is very well written and the data are presented clearly. The issue investigated is clinically relevant and the conclusions are clear and definitive - no benefit was observed with combination versus monotherapy. In addition to answering this primary research question, the results provide further evidence that ACE inhibitor therapy affords similar benefit to angiotensin receptor blocker therapy. 

Comments:

* Page 5, rows 109-110: This sentence is a little unclear and could be shortened to: "Secondarily, we compared the protective effect against microalbuminuria development of two monotherapies in the same population". 

* Page 10, rows 216-217: The authors should provide a more detailed explanation of "acceleration factors" since these are not commonly reported.

* Table 1: A footnote states that measured GFR was obtained in a subgroup of participants. It would be helpful also to provide the number (n=77)

* Measured GFR (plasma iohexol clearance) was assessed in only a minority of participants but serum creatinine was measured in all. The authors should therefore report estimated GFR values for all participants in Table 1. Also, estimated GFR could be used to calculate the slope of GFR over time in all participants.

***

[LINK]

---

## [Editor Report · Decision Letter 2]

4 Jun 2021

Dear Dr. Cortinovis,

Thank you very much for re-submitting your manuscript "Preventing microalbuminuria with benazepril, valsartan and benazepril-valsartan combination therapy in diabetic patients with high-normal albuminuria: a prospective, randomised, open label, blinded endpoint (PROBE) study" (PMEDICINE-D-21-01663R2) for consideration at PLOS Medicine.

I have discussed the paper with editorial colleagues and our academic editor, and I am pleased to tell you that, provided the remaining editorial and production issues are fully dealt with, we expect to be able to accept the paper for publication in the journal.

[LINK]

Please let me know if you have any questions, and we look forward to receiving the revised manuscript shortly.   

Sincerely,

Richard Turner, PhD

rturner@plos.org

Requests from Editors:

Please remove the time restraints from your data statement (submission form) - so as to comply with PLOS' data policy, data must be available in principle from the date of publication.

At lines 52 and 171 there is a slight difference in the way the participating units are described. 

At line 294, please make that "Two hundred ...".

Throughout the text, please use the style "... 3 study groups ...", although numbers should be spelt out at the start of sentences. 

Currently, the Acknowledgements section appears twice at the end of the ms.

Comments from Academic Editor:

In my view the authors have responded adequately to all reviewer comments. This is an important paper as it represents another (and possibly the final) milestone in the long-running exploration of whether inhibition of the renin-angiotensin system with ACE inhibitors or angiotensin receptor blockers or their combination offers superior renoprotective benefits (see Taal MW and Brenner BM. Renoprotective benefits of RAS inhibition: from ACEI to angiotensin II antagonists. Kidney Int. 2000 May;57(5):1803-17). There was a compelling rationale for combining ACE inhibitor and angiotensin receptor blocker therapy but clinical trials have not shown renoprotective benefit in different types of kidney disease.

I agree with the authors that comparison of blood pressure control between groups should be reported in the abstract as this is vital for interpretation of the results.

There are a few very minor language issues that could be resolved during production or during a minor revision. I therefore recommend minor revision with a view to accepting this paper for publication.

Comments:

Line 82-83: suggest rephrase to “Treatment effect was, however, associated with greater blood pressure reduction.”

Line 162-163: suggest rephrase to “This was a phase 3, multicenter, parallel PROBE trial [14], that included male and female patients aged over 40 years with type 2 diabetes (WHO criteria) and a known history of diabetes not exceeding 25 years.”

Line 172: word missing, suggest change to “…excluded patients with drug or alcohol abuse…”

Line 243: suggest change to “…who was not directly involved in conducting the study.”

Line 360-362: suggest change to “Non-pre-specified analyses considered in addition to MACE, included coronary or peripheral revascularization or amputation because of critically ischemic limb.”

***

---

## [Editor Report · Decision Letter 3]

10 Jun 2021

Dear Dr Cortinovis, 

On behalf of my colleagues and the Academic Editor, Dr Taal, I am pleased to inform you that we have agreed to publish your manuscript "Preventing microalbuminuria with benazepril, valsartan and benazepril-valsartan combination therapy in diabetic patients with high-normal albuminuria: a prospective, randomised, open label, blinded endpoint (PROBE) study" (PMEDICINE-D-21-01663R3) in PLOS Medicine.

Prior to final acceptance we suggest some small changes to wording in the abstract: at line 60 "Between-group differences in estimated acceleration factors were non-significant after adjustment ..."; at line 63, would that be lower than expected "prevalence of albuminuria"?

PRESS

Sincerely, 

Richard Turner, PhD 

rturner@plos.org